# Unimolecular Micelles from Randomly Grafted Arborescent Copolymers with Different Core Branching Densities: Encapsulation of Doxorubicin and In Vitro Release Study

**DOI:** 10.3390/ma16062461

**Published:** 2023-03-20

**Authors:** Mosa Alsehli, Mario Gauthier

**Affiliations:** 1Department of Chemistry, Taibah University, Medina P.O. Box 344, Saudi Arabia; 2Department of Chemistry, Institute for Polymer Research, University of Waterloo, 200 University Ave. W., Waterloo, ON N2L 3G1, Canada

**Keywords:** doxorubicin, arborescent poly(benzyl L-glutamate), drug delivery, sustained release

## Abstract

A series of amphiphilic arborescent copolymers of generations G1 and G2 with an arborescent poly(γ-benzyl L-glutamate) (PBG) core and poly(ethylene oxide) (PEO) chain segments in the shell, PBG-*g*-PEO, were synthesized and evaluated as drug delivery nanocarriers. The PBG building blocks were generated by ring-opening polymerization of γ-benzyl L-glutamic acid *N-*carboxyanhydride (Glu-NCA) initiated with *n*-hexylamine. Partial or full deprotection of the benzyl ester groups followed by coupling with PBG chains yielded a comb-branched (arborescent polymer generation zero or G0) PBG structure. Additional cycles of deprotection and grafting provided G1 and G2 arborescent polypeptides. Side chains of poly(ethylene oxide) were then randomly grafted onto the arborescent PBG substrates to produce amphiphilic arborescent copolymers. Control over the branching density of G0PBG was investigated by varying the length and the deprotection level of the linear PBG substrates used in their synthesis. Three G0PBG cores with different branching densities, varying from a compact and dense to a loose and more porous structure, were thus synthesized. These amphiphilic copolymers behaved similar to unimolecular micelles in aqueous solutions, with a unimodal number- and volume-weighted size distributions in dynamic light scattering measurements. It was demonstrated that these biocompatible copolymers can encapsulate hydrophobic drugs such as doxorubicin (DOX) within their hydrophobic core with drug loading efficiencies of 42–65%. Sustained and pH-responsive DOX release was observed from the unimolecular micelles, which suggests that they could be useful as drug nanocarriers for cancer therapy.

## 1. Introduction

Amphiphilic block copolymer micelles have attracted much attention for biomedical applications such as drug delivery [1,2,3]. One of the most important characteristics of polymeric micelles in drug delivery applications is their ability to solubilize hydrophobic or poorly water-soluble drugs within their core, and thus enhance their bioavailability [4,5]. Polymeric micelles obtained by the self-assembly of linear block copolymers are always at dynamic equilibrium with non-associated (unimer) polymer chains. The self-assembly of isolated polymer chains to form micelles only starts when a minimum concentration is reached, known as the critical micelle concentration (CMC) [6]. As a consequence, when applied in vivo, polymeric micelles based on linear block copolymers face a difficulty related to their dilution in the body, which shifts the micellar equilibrium toward the unimer state and causes drug leaching, thus reducing the efficacy of sustained drug delivery [7,8,9,10].

Dendritic polymer micelles are an interesting alternative to conventional amphiphilic block copolymer micelles, in that each micelle is formed by a single molecule with amphiphilic properties (unimolecular micelles), with a covalently bound core-shell structure [11]. In contrast, to block copolymer micelles, unimolecular micelles are stable irrespective of their concentration or the solvency conditions [9,12,13,14,15,16,17]. Due to these remarkable features, the past decade has witnessed a surge of interest in the synthesis and characterization of unimolecular micelles, particularly in relation to biomedical applications. To date, the polymers most widely investigated to form unimolecular micelles for various applications are dendrimers and hyperbranched polymers. Unfortunately, dendrimers are time-consuming and tedious to synthesize, while hyperbranched polymers are easier to synthesize but only with limited control over their structure [18,19,20].

In contrast to dendrimers and hyperbranched polymers, dendrigraft (arborescent) polymers are a subset of dendritic polymers derived from reactive polymer building blocks in successive grafting reactions rather than small molecule monomers [21,22]. Due to the presence of a large number of coupling sites distributed randomly on the grafting substrates, a rapid increase in molecular weight is observed over successive arborescent polymer generations and high molecular weight species are obtained in a few steps, while narrow molecular weight distributions (M_w_/M_n_ < 1.1) are maintained in most cases [23]. The ability of nanocarriers to accumulate in target tissues depends on their physicochemical characteristics, among which the particle size is critical. The formulation of safe, stable, and efficient drug delivery systems, with well-defined and predictable release characteristics, requires nanocarriers with a uniform (close to monodisperse) size population [24]. Arborescent unimolecular micelles can be obtained by the addition of hydrophilic segments in the last grafting cycle. For example, arborescent polystyrene-*graft*-poly(2-vinylpyridine) (PS-*g*-P2VP) copolymers were synthesized by grafting poly(2-vinylpyridine) side chains onto arborescent PS substrates of different generations. These copolymers were freely soluble in acidic aqueous solutions, behaving similar to unimolecular micelles that were able to solubilize various polycyclic aromatic hydrocarbon probes within their hydrophobic core [25]. In a subsequent study, the release properties of these systems loaded with indomethacin and lidocaine as model drugs were also investigated [26]. Unfortunately, the arborescent PS-*g*-P2VP copolymers used in the investigation were not biocompatible and only water-soluble at low pH due to protonation of the P2VP side chains, and thus the conditions of these in vitro investigations were very different from real biological systems. The synthesis of fully biocompatible arborescent polymers from poly(γ-benzyl L-glutamate) (PBG) side chains represents a significant advancement in that area [27]. Our group previously reported the synthesis of arborescent PBG substrates grafted either randomly or at their chain ends with linear poly(ethylene oxide), polyglycidol, and poly(glutamic acid) to form a hydrophilic shell [28].

Despite the remarkable potential of arborescent micelles as drug nanocarriers, the use of fully biocompatible arborescent copolymers in drug delivery has yet to be explored. We herein expand the design and synthesis of biocompatible amphiphilic arborescent copolymers as drug nanocarriers. Arborescent copolymers with poly(γ-benzyl L-glutamate) (PBG) cores of generations G1 and G2, derived from G0PBG substrates with different branching densities, and a shell of poly(ethylene oxide) (PEO) chain segments were examined. The effects of the PBG core branching density and generation number on the micelle size, drug loading capacity, and drug release rate were investigated. The copolymers were characterized in solution by dynamic light scattering (DLS), proton nuclear magnetic resonance (^1^H NMR), size exclusion chromatography (SEC) measurements, and in the solid state with transmission electron microscopy (TEM), and atomic force microscopy (AFM). The encapsulation and release properties were investigated by UV spectroscopy using doxorubicin (DOX) as a model hydrophobic drug.

## 2. Materials and Methods

### 2.1. Materials

Dimethyl sulfoxide and *n*-hexylamine were purified by stirring overnight with CaH_2_ and distillation under reduced pressure. *N,N*-Dimethylformamide serving in the polymer synthesis (DMF; Aldrich, Oakville, ON, Canada, >99%) was purified by distillation under reduced pressure and stored in the dark [27,28]. Ethyl acetate (Fisher, Ottawa, ON, Canada, 99.9%) was distilled from LiAlH_4_ under nitrogen. γ-Benzyl L-glutamic acid (Bz-Glu; Bachem, Torrance, CA, USA, >99%), diethyl ether (EMD Millipore, Oakville, ON, Canada, OmniSolv), HBr solution (Aldrich, 33% in acetic acid), *N,N*’-diisopropylcarbodiimide (DIC; Aldrich, 99%), trifluoroacetic acid (TFA, Caledon, Georgetown, ON, Canada, 99.9%), 1-ethyl-3-(3-dimethylaminopropyl)carbodiimide hydrochloride (EDC·HCl; Aldrich, 98%), 1-hydroxybenzotriazole (HOBt; Fluka, water content ca. 15% *w*/*w*), tetrahydrofuran (THF, EMD Millipore OmniSolv), methanol (EMD Millipore OmniSolv), triphosgene (Aldrich, 98%), LiAlH_4_ (Aldrich, 95%), acetic anhydride (Caledon, 97%), deuterated DMSO (DMSO-*d_6_*, Cambridge isotopes, Andover, MA, USA, 99.9% D), deuterated H_2_O (D_2_O; Aldrich, 99.9 atom % D), and triethylamine (TEA, EMD) were used as received from the suppliers. Doxorubicin hydrochloride (DOX·HCl) was obtained from Sigma-Aldrich (St. Louis, MO, USA). Dialysis tubes Spectra/Por^®^ 7 (MWCO 3.5 kDa) were purchased from Spectrum Laboratories Inc. (New Brunswick, NJ, USA).

### 2.2. Characterization

Proton nuclear magnetic resonance (^1^H NMR) spectroscopy served to estimate the degree of polymerization of the linear poly(γ-benzyl L-glutamate) PBG chains, to determine the deprotection level of the PBG substrates, and to investigate the ability of randomly grafted arborescent copolymers to form micelles. The instrument used was a Bruker 300 MHz spectrometer (Bruker, Milton, ON, Canada). Fluorine nuclear magnetic resonance (^19^F NMR) spectroscopy was also performed on the Bruker 300 MHz instrument to determine the chain end primary amine functionality, *f*_NH2_, of the polymers used in the grafting reactions, by a procedure developed by Ji et al. [29].

Analytical SEC, for the characterization of the arborescent PBG substrates of generations G0-G2 and of the copolymers, was carried out on a Viscotek GPCmax unit (Malvern Panalytical, Westborough, MA, USA) equipped with a VE 2001 SEC Solvent/Sample Module, a Viscotek double detector array with refractive index and dual-angle light scattering detectors, and two Agilent Technologies (Mississauga, ON, Canada) PLgel 5 µm MIXED-C and PLgel 10 µm MIXED-B organic mixed bed columns, with dimensions of 7.5 mm (ID) × 300 mm (L). The system was operated at a flow rate of 0.5 mL/min at 70 °C, using dimethyl sulfoxide (DMSO) with LiBr (0.05 M) as the mobile phase. Analysis of the chromatograms was performed with the OmniSEC 4.6.1 software package.

Preparative SEC was conducted on a system with a Waters (Mississauga, ON, Canada) M45 HPLC pump, a 2-mL sample injection loop, a Waters R401 differential refractometer detector, and a Jordi Gel DVB 1000 Å 250 mm × 22 mm preparative SEC column (Jordi Labs, Mansfield, MA). *N,N*-Dimethylformamide (DMF) with 0.2 g/L LiCl served as the mobile phase. The concentration of polymer injected was 25 mg/mL. The system was operated at a flow rate of 3 mL/min at room temperature (25 °C).

Light scattering measurements were performed on a Zetasizer Nano ZS90 (Malvern Panalytical) equipped with a 4 mW He–Ne laser operating at 633 nm and 25.0 °C, at a scattering angle of 90°. The samples were dissolved in phosphate-buffered saline (PBS), DMF, or water for at least 12 h before analysis.

Absorption spectra were obtained on a Cary 100 Bio UV-Vis spectrophotometer (Agilent) with a spectral bandwidth (SBW) of 2 nm, operated with the Cary Varian UV Scan Application (v3.001339). The absorption peak at 483 nm for doxorubicin and a molar extinction coefficient ε = 10,240 M^−1^·cm^−1^ was used to calculate the doxorubicin loading in the DOX-micelle samples.

The dendritic micelles were imaged with a Philips CM10 electron microscope operated at 60 kV. Samples for the TEM measurements were prepared by the following method: One drop of solution (0.05 mg∙mL^−1^) was cast onto a 300-mesh Formvar^®^ carbon-coated copper TEM grid placed on filter paper and the excess solution was wicked off with filter paper. After 1 min, one drop of 2% (*w*/*v*) phosphotungstic acid was added to the grid and the excess staining solution was wicked off with filter paper. Finally, the grid was transferred to a new piece of filter paper in a Petri dish and left to dry overnight at room temperature.

Atomic force microscopy images were recorded in the tapping mode on a Nanoscope III instrument (Veeco, Plainview, NY, USA, model MMAFM-2, scan stage J). The polymer solutions were prepared at concentrations ranging from 0.01 to 0.05 mg.mL^−1^. A 20 μL aliquot of the solution was deposited on the mica substrate and spun at about 3000 revolutions per minute (rpm) for 60 s under ambient conditions. The images were analyzed using the Nanoscope v 1.40 software. The scan rate was typically between 0.7 and 1.2 Hz, at a scan angle of 0°, acquiring 512 samples/line.

### 2.3. Synthetic Procedures

The monomer γ-benzyl L-glutamic acid *N-*carboxyanhydride (Glu-NCA) was synthesized from the corresponding *α*-amino acid by refluxing with triphosgene in ethyl acetate under nitrogen as described in the literature [30]. The synthesis of the poly(γ-benzyl L-glutamate) PBG side chains, three linear PBG substrates with either 15, 30, or 68 repeating units, the partially or fully deprotected linear PBG substrates, and linear poly(ethylene oxide) (PEO) were accomplished by ring-opening polymerization as described previously [27]. Detailed information on the synthesis and the characterization is provided as Appendix A. The different linear PBG substrates are identified in the form PBG_x_, where X denotes the experimental number of BG repeating units in the chains. The arborescent G0PBG molecules obtained by coupling the amine-terminated PBG side chains with the different carboxyl-functionalized PBG substrates are likewise identified in the form G0PBG_x,_ where X denotes the number of BG repeating units in the substrate used for the synthesis of G0PBG. For example, G0PBG_29_ describes an arborescent polymer derived from the PBG substrate with X_n_ = 29. Similar sample notation was used for the G1 and G2 samples. The yield of the polymerization reactions for the linear PBG substrates was 89–92% (see Appendix A).

#### 2.3.1. Synthesis of G0 Arborescent PBG

The synthesis of arborescent G0PBG was achieved using optimized coupling reaction conditions [31]. In the reactions used for the synthesis of G0PBG_29_ and G0BPG_65_, the partially (31%) deprotected polymer serving as substrate (1.10 g, 1.78 mmol -CO_2_H) and the polymer serving as side chains (5.39 g, 1.05 mmol chains) were dissolved in 50 mL of dry DMSO. The peptide coupling reagents DIC (2.37 mL, 15.2 mmol) and HOBt (2.05 g, 15.2 mmol) were then added to the reaction with TEA (0.73 mL, 5.2 mmol). The reaction was allowed to proceed for 36 h at room temperature before adding *n*-hexylamine (1.84 mL, 18.2 mmol) to deactivate residual activated carboxylic acid sites. After 3 h, the product was precipitated in cold methanol and recovered by suction filtration. Unreacted side chains were removed from the G0 crude polymer by preparative size exclusion chromatography (SEC).

In the coupling reaction for G0PBG_15_, the fully deprotected polymer serving as substrate (0.14 g, 1.08 mmol -CO_2_H) and the polymer serving as side chains (5.07 g, 0.98 mmol chains) were dissolved in 45 mL of dry DMSO. The peptide coupling reagents DIC (0.93 mL, 6.0 mmol) and HOBt (0.81 g, 6.0 mmol) were then added to the reaction with TEA (0.69 mL, 4.9 mmol). The reaction was allowed to proceed for 36 h at room temperature before adding *n*-hexylamine (0.72 mL, 7.2 mmol) to deactivate residual activated carboxylic acid sites. After 3 h, the product was precipitated in cold methanol and recovered by suction filtration. Unreacted side chains were removed from the G0 crude polymer by preparative size exclusion chromatography (SEC).

The procedures used for the synthesis of the G1 and G2 PBG substrates and characterization information (including the grafting yield) are provided as Appendix A.

#### 2.3.2. Synthesis of Arborescent Copolymers

The arborescent copolymers were synthesized and purified similarly to the arborescent PBG samples. The synthesis of G1PBG_15_-*g*-PEO is described below as an example. The partially deprotected G1PBG_15_ substrate (30 mol % free glutamic acid moieties, 0.040 g, 0.063 mmol –CO_2_H) and amine-terminated PEO (M_n_ = 10,100 g/mol) serving as side chains (0.58 g, 0.057 mmol chains) were dissolved in 5 mL of dry DMSO. The peptide coupling reagents DIC (0.055 mL, 0.35 mmol) and HOBt (0.047 g, 0.35 mmol) were then added to the reaction with TEA (0.040 mL, 0.29 mmol). The reaction was allowed to proceed for 36 h at room temperature before adding *n*-hexylamine (0.042 mL, 0.42 mmol) to deactivate residual activated carboxylic acid sites. After 3 h, the product was precipitated in cold methanol and recovered by suction filtration. Unreacted side chains were removed from the G1PBG_15_-*g*-PEO crude product by preparative size exclusion chromatography (SEC) in DMF and the sample was recovered by precipitation in cold diethyl ether, suction filtration, and drying under vacuum: Grafting yield = 27%, M_n_ = 1.2 × 10^6^, M_w_/M_n_ = 1.07 (MALLS).

### 2.4. Loading of DOX in the Unimolecular Micelles

The as-supplied DOX·HCl was combined with two equivalents of triethylamine (TEA) in DMSO to obtain the drug its free base (hydrophobic) form. The arborescent copolymer (10 mg) was dissolved in 1 mL of DMSO and stirred for 2 h. Then 0.5 mL of DOX solution in DMSO (4 mg/mL) was added to the micellar solution and the mixture was stirred overnight in the dark. The organic solvent and free drug were removed by dialysis (MWCO 3500) against deionized water (1 L) for 24 h (the dialysis medium was changed three times) and the solution was either lyophilized in the dark or used directly for the measurements. For the determination of the drug loading content (DLC) and the drug loading efficiency (DLE), the lyophilized DOX-loaded micelles were dissolved in deionized water and the absorbance was measured on a UV-Vis spectrometer at 483 nm. The DLC and DLE were calculated according to the following equations:(1)$$DLC=\frac{mass\;of\;drug\;in\;micelles}{mass\;of\;micelles\;and\;drug}\;\times \;100\%$$
(2)$$DLE=\frac{mass\;of\;drug\;in\;micelles}{total\;mass\;of\;drug\;in\;feed}\;\times \;100\%$$

### 2.5. In Vitro Release of DOX

To determine the DOX release profiles, a 3 mg sample of the freeze-dried DOX-loaded micelles was suspended in 1 mL of phosphate-buffered saline (release medium, 10 mM sodium phosphate, 137 mM NaCl, and 2.7 mM KCl, pH 7.4 or 5.5; the pH was adjusted by addition of 6 M HCl) and transferred into a dialysis bag (MWCO 3500). The release experiment was initiated by placing the sealed dialysis bag into 3 mL of release medium at the same pH and 37 °C, with constant shaking at 100 rpm. At selected time intervals, the release medium was completely withdrawn and replaced with 3 mL of fresh medium. The amount of DOX released was calculated based on the absorbance measured on a UV-Vis spectrometer at 483 nm and a molar extinction coefficient ε = 10,240 M^−1^·cm^-1^. The drug release studies were performed in triplicate for each sample.

## 3. Results and Discussion

### 3.1. Synthesis of Linear PBG Substrates

The characteristics of arborescent copolymers can be tailored to meet specific requirements for different applications through variations in the synthetic procedure. For instance, the molecular weight of the linear substrate serving in the synthesis of the G0 polymer, the functionalization level of the substrate, and the number of grafting cycles used in their synthesis can be changed. In the current investigation, the length and the deprotection level of the linear PBG substrate used in the synthesis of G0PBG were varied, while the length of the PBG side chains (core building blocks) had M_n_ ≈ 5000 g/mol in all cases. Linear PBG substrates with three different lengths were synthesized with X_n_ = 15, 29, and 65, as shown in Appendix A. The number of benzyl glutamate repeating units were controlled through the ratio of monomer (Glu-NCA) to the initiator (*n*-hexylamine) used in the polymerization reaction. Linear PBG substrates with different X_n_ and deprotection levels were employed in the synthesis of G0PBG in the hope that the shape of the molecules could be controlled from spherical to brush-like topologies. The shape of micelles can significantly alter their behavior in living organisms and cells. For example, it was reported that worm-like polymer micelles of amphiphilic poly(ethylene glycol)-*block*-polycaprolactone copolymers had an increased circulation time in comparison with their spherical counterparts [32,33]. Unfortunately, TEM and AFM imaging indicated that the molecules were all spherical. This approach nevertheless provided control over the branching density in the cores, which seems to correlate very well with the release properties of the micelles, as discussed in the release study. 

As shown in Figure 1, absolute values of M_n_ and X_n_ were determined from ^1^H NMR analysis of the linear PBG samples, by comparing the integrated peak intensities for the benzylic methylene protons in the repeating units (4.9 ppm) and the terminal methyl group signal from the *n*-hexylamine initiator (0.77 ppm).

The branching functionality (*f*_n_) of arborescent polymer molecules can be manipulated through the functionalization (deprotection) level of the substrate. In the current investigation, only the deprotection level of the linear PBG substrate serving in the G0 polymer synthesis was varied (Appendix A). The treatment of linear PBG with HBr in TFA allowed the cleavage of a controlled fraction of benzyl ester-protecting groups to generate coupling sites (carboxyl groups) for the grafting reaction. A target deprotection level of 30% was used for PBG_29_ and PBG_65_, while 100% deprotection was used for PBG_15_ (Appendix A). The deprotection level was controlled through the amount of HBr added with respect to the benzyl ester moieties. The ^1^H NMR spectra for partly deprotected PBG_29_ and PBG_65_, and fully deprotected PBG_15_ are compared in Appendix A. The ratio of integrated intensities for the benzylic methylene protons (2H at 4.9 ppm) for the remaining protected structural units to the methine protons (1H at 3.9 ppm) served to determine the deprotection level of PBG_29_ and PBG_65_.

### 3.2. Synthesis of G0PBG with Different Branching Densities

The synthesis of arborescent G0PBG from amine-terminated PBG side chains and carboxyl-functionalized PBG substrates of different lengths and deprotection levels was achieved under previously optimized coupling reaction conditions [31]. The coupling reagents *N,N*′-diisopropylcarbodiimide (DIC) and 1-hydroxybenzotriazole (HOBt) were used to activate the carboxyl groups. All the reactions were carried out in DMSO at 25 °C, using a 5:1 molar ratio of coupling agents (DIC/HOBt) to CO_2_H groups on the substrate to maximize the grafting yield and the coupling efficiency. These optimized conditions were also subsequently used to synthesize the G1 and G2 arborescent PBG. The general coupling reaction for the synthesis of arborescent PBG is shown in Figure 1, using a G0 polymer synthesis as an example.

Different chain lengths and deprotection levels for the linear PBG substrate were used in the synthesis of G0PBG, while the PBG side chains had M_n_ ≈ 5000 g/mol in all cases. Three well-defined (dispersity Ð = M_w_/M_n_ ≤ 1.08) arborescent G0 (comb-branched) PBG were obtained in that manner: G0PBG_15_ from a PBG substrate with X_n_ = 15 and 100% deprotection, G0PBG_29_ from a PBG substrate with X_n_ = 29 and 31% deprotection, and G0PBG_65_ from a PBG substrate with X_n_ = 65 and 31% deprotection. The corresponding characterization data are summarized in Table 1.

As shown in Table 1, the number-average branching functionality of the arborescent polypeptides, *f*_n_, defined as the number of side chains added in the grafting reaction, increased with the deprotection level since more coupling sites were available. However, despite the identical deprotection levels of PBG_29_ and PBG_65_, the branching functionality *f*_n_ achieved for G0PBG_29_ was marginally lower. This is attributed to increased steric crowding leading to less accessible carboxylic acid moieties in the shorter PBG_29_ substrate. The number-average branching density (*b*_d_), defined as the number of side chains added in the grafting reaction divided by the number of repeating units in the linear substrate, clearly increased as the chain length of the substrate decreased. As a result, a compact and dense structure is expected for G0PBG_15_ (*b*_d_ = 0.80), as opposed to a loose and more porous structure for G0PBG_65_ (*b*_d_ = 0.11), and an intermediate (semi-compact) structure for G0PBG_29_ (*b*_d_ = 0.19). The synthesis of the arborescent G0PBG samples with different branching densities (porosities) is illustrated with their SEC elution curves in Appendix A.

The G1 and G2 PBG arborescent substrates were synthesized and purified by the same method used for the G0 samples except that, in all cases, the deprotection level of the arborescent substrates used was ≈30% and a 1:1.1 molar ratio of side chains (PBG-NH_2_) to CO_2_H groups was used to maximize the grafting yield and the coupling efficiency (Appendix A). The molecular weight of the polymers increased with the generation number, while a low dispersity (Ð ≤ 1.09) was maintained. The branching functionality also increased over successive generations, as more coupling sites were available after each grafting cycle. Sample G1PBG_15_ showed the lowest G_y_ and C_e_, which is attributed to the difficulty for the coupling sites to react in the compact and dense G0PBG_15_ substrate because of increased crowding.

### 3.3. Synthesis of Randomly Grafted Arborescent Copolymers

Structural variables such as the size and the branching density (porosity) of arborescent copolymers can be designed to meet requirements for specific applications. Whitton and Gauthier previously reported the use of linear poly(ethylene oxide) with a molecular weight of 5000 (PEG_110_), grafted either randomly or at the end of the chains on arborescent PBG substrates to generate arborescent copolymer micelles [28]. Unfortunately, the randomly grafted arborescent G1PBG-*g*-PEO_110_ copolymers obtained by that approach yielded large aggregated species in phosphate-buffered saline (PBS), while G2PBG-*g*-PEO_110_ and G3PBG-*g*-PEO_110_ were both insoluble. The corresponding end-grafted samples PBG-*e*-PEO_110_ displayed better dispersibility in PBS than the randomly grafted systems, albeit a small population of aggregates still existed in these samples. Therefore, we hypothesized that by randomly grafting longer hydrophilic PEO chains (M_n_ = 10,000 g/mol), the solubility of the copolymers in PBS would be enhanced to yield stable unimolecular micelles with minimal aggregation. This would in turn affect the size and the properties of the micelles, and thereby provide control over the encapsulation and release behaviors of these unimolecular micelles. A detailed discussion of the synthesis of linear amine-terminated PEO, characterization results, and ^19^F NMR analysis to determine the amine functionality level of PEO (*f*_NH2_) is provided in the SM.

The synthesis of amphiphilic arborescent copolymers from amine-terminated PEO chains and the randomly carboxyl-functionalized PBG substrates was achieved under previously optimized reaction conditions [28]. The coupling of a G1PBG substrate with amine-terminated PEO chains is illustrated in Figure 2 as an example of the synthesis of an amphiphilic arborescent copolymer.

The results obtained for grafting PBG substrates of generations G1 and G2 with PEO side chains are summarized in Table 2. The absolute molecular weight of the substrates and the arborescent copolymers was determined by SEC-MALLS analysis in DMSO. In all cases, the molecular weight of the copolymers increased relatively to the PBG substrates, while the dispersity remained low (Ð ≤ 1.09).

The grafting yield (G_y_), defined as the fraction of linear chain segments becoming attached to the substrate, was determined from the weight fraction of each component in the copolymers, along with the known amounts of substrate and side chains used in each grafting reaction. The calculation of the grafting yield for sample G1PBG_15_-*g*-PEO is provided in Equation (S2) (Appendix A) as an example. Several factors could have contributed to the relatively low (13–29%) grafting yields in the synthesis of the amphiphilic arborescent copolymers. It was assumed that partial deprotection of PBG would yield randomly distributed coupling sites within the substrates, but the deprotection of adjacent groups is also possible, which may lead to increased coupling site crowding. The grafting yield achieved for the G1PBG-*g*-PEO copolymers was higher than for the G2PBG-*g*-PEO copolymers, which is also expected due to the increased rigidity of PBG substrates of higher generations.

The branching functionality *f*_n_, defined as the number of PEO side chains added, was determined by dividing the increase in molecular weight observed for the grafted copolymer by the molecular weight of the PEO side chains. A calculation for sample G1PBG_15_-*g*-PEO is provided in Equation (S3) as an example. All the copolymers synthesized were purified by preparative SEC. The SEC traces for the purified samples, corresponding to the data summarized in Table 2, are compared in Figure 2. As expected, a decrease in elution volume was observed as the generation number increased. Within each generation, the elution volume decreased in the series PBG_65_-*g*-PEO < PBG_15_-*g*-PEO < PBG_29_-*g*-PEO. PBG_65_-*g*-PEO had the highest molecular weight, likely due to its more porous core structure minimizing steric crowding and facilitating the diffusion of the chains to the coupling sites in the grafting reaction.

#### Properties of the Arborescent Copolymer Micelles

The arborescent copolymers were evaluated for their potential to remain as unimolecular (non-aggregated) micelles in aqueous PBS solutions. The use of ^1^H NMR spectroscopy to investigate the formation of micellar structures is illustrated in Appendix A (SM). Complete disappearance of the peak’s characteristic for PBG such as the benzylic methylene (2H at 4.9 ppm) and phenyl protons (5H at 7.3 ppm) was observed in D_2_O, while these peaks were clearly visible in DMSO-*d_6_*, which confirms that G2PBG-*g*-PEO has a core–shell morphology in aqueous solutions with a collapsed hydrophobic PBG core and a hydrophilic PEO shell.

The hydrodynamic diameter of the copolymers, determined by DLS analysis in DMF (good solvent for PBG and PEO) and in aqueous PBS buffer (solvent selective for PEO) followed the expected trends for increasing generation numbers (Appendix A). The good agreement for the hydrodynamic diameters obtained in DMF and PBS confirms that the randomly grafted arborescent copolymers formed unimolecular micelles, with mean number-average diameters between 16–18 nm and 32–37 nm for the G1 and G2 in DMF, respectively. As shown in Appendix A, the unimolecular micelles are uniform in size since the values of the three average diameters (number, volume, and intensity) are similar. The copolymers should form unimolecular, non-aggregated solutions in DMF, a good solvent for both the PBG and PEO components. Analysis of the same copolymers in phosphate-buffered saline (PBS) also yielded values for number-, volume-, and intensity-average diameters similar to those obtained in DMF, which suggests that all the arborescent copolymers also existed as unimolecular species under these conditions. It is nonetheless clear that these values represent differently weighted means of the size distributions, and indeed size distribution analysis (Figure 3) shows that all the G1PBG-*g*-PEO copolymers exhibit a small quantity (16–34%) of aggregated structures with a diameter of 70–160 nm in their *intensity* distributions, while no aggregation is detected in neither the *number* nor the *volume* distributions. Size distribution analysis of the G2PBG-*g*-PEO copolymers likewise yielded a small quantity (17–32%) of aggregated structures with a diameter of 160–230 nm in the *intensity* distributions, while no aggregation was detected in neither the *number* nor the *volume* distributions (Appendix A). These results indicate that even though the vast majority of randomly grafted arborescent copolymer molecules exist as stable, isolated unimolecular micelles, a minute amount of aggregation occurs that is only detectable in the intensity-weighted distributions, due to the extremely high sensitivity of the scattering intensity to large particles [34]. Arborescent copolymers of the same generation yielded similar hydrodynamic diameters by DLS analysis even though their core structure was different. For example, when comparing the micelles formed by G2PBG_15_-*g*-PEO and G2PBG_65_-*g*-PEO, there is only a 4 nm difference in the number-average diameter, confirming that the core only has a minor influence on the overall micelle size. Since the hydrophobic PBG cores should collapse to minimize their unfavorable interactions with the aqueous solution (as determined by ^1^H NMR analysis) and the G1 and G2 cores have a similar molecular weight in each series, their contribution to the overall size of the micelles should be minimized in comparison to the more soluble PEO shell.

The morphology and size of the unimolecular micelles were investigated by TEM, after staining with phosphotungstic acid on a carbon-coated copper grid, as well as by AFM (Appendix A for the G1 copolymers, Figure 4 for the G2 copolymers). The unimolecular micelles have a fairly uniform size distribution and a spherical shape in all cases. The average micelle diameter measured by TEM was 15 ± 2, 14 ± 3, and 16 ± 2 nm for G1PBG_15_-*g*-PEO, G1PBG_29_-*g*-PEO, and G1PBG_65_-*g*-PEO, respectively. In contrast, the average micelle diameter was 29 ± 3, 30 ± 2, and 35 ± 3 nm for G2PBG_15_-*g*-PEO, G2PBG_29_-*g*-PEO, and G2PBG_65_-*g*-PEO, respectively. All the measurements were based on 15–30 particles within the same micrograph. The TEM measurements confirmed the trends in the DLS measurements in terms of increasing micelle size for higher generations. The diameter of these unimolecular micelles appears to be within the ideal size range for therapeutic nanocarriers to be used in cancer treatment [35]. The size of the micelles measured by TEM was somewhat smaller than by DLS analysis. Because TEM yields the diameter of the dry, somewhat flattened micelles on the TEM grid surface, while DLS measures their hydrodynamic (solvated) diameter, discrepancies are to be expected between these methods. The results obtained are therefore considered to be in good agreement.

### 3.4. Drug Loading and Micelle Characterization

Since the arborescent PBG-*g*-PEO copolymers have a hydrophobic PBG core and a hydrophilic PEG shell, these amphiphilic molecules could be useful as nanocarriers to physically encapsulate hydrophobic guest molecules through hydrophobic interactions (Figure 3).

The ability to load the hydrophobic core of these unimolecular micelles with guest molecules was investigated to determine whether the differences in their core structure affected their ability to encapsulate and control the rate of release of the hydrophobic drug. Doxorubicin (DOX), an anticancer drug, was selected as a model drug to assess loading and release in vitro. DOX·HCl was treated with triethylamine before encapsulation to obtain DOX in its free base (more hydrophobic) form. DOX was encapsulated within the hydrophobic core of G1PBG-*g*-PEO and G2PBG-*g*-PEO by co-dissolution followed by dialysis. It was indeed determined that hydrophobic DOX could be solubilized in the amphiphilic arborescent copolymers, as shown in Figure 5. For example, the successful encapsulation of DOX within the hydrophobic core of G2PBG-*g*-PEO is indicated by the enhanced absorption around 490 nm of the DOX-loaded micellar solutions (G2PBG-*g*-PEO/DOX) as compared to the empty micelles (G2PBG-*g*-PEO). The encapsulation of DOX specifically within the hydrophobic core of the micelles is also suggested by the blue shift in the maximum absorption for G2PBG-*g*-PEO/DOX as compared to free DOX in PBS (λ_max_ = 483 nm), which is attributed to the hydrophobic microenvironment created by the PBG core of G2PBG-*g*-PEO [36].

The existence of hydrophobic interactions was further confirmed by ^1^H NMR analysis of the G2PBG_15_-*g*-PEO/DOX unimolecular micelles in D_2_O (Figure 6). The complete disappearance of both the PBG and DOX signals for G2PBG_15_-*g*-PEO/DOX in D_2_O (Figure 6c) in comparison to the characteristic signals observed of free DOX in D_2_O at the same overall DOX concentration (Figure 6a) confirms the restricted mobility of both the PBG and DOX components within the core of G2PBG_15_-*g*-PEO. This again supports the solubilization and encapsulation of DOX molecules within the core of the micelles in aqueous solutions.

The drug loading content (DLC) determined for the micelles ranged from 7.1–10.8 wt%, and the drug loading efficiency (DLE) from 42–65% for the different G1PBG and G2PBG copolymers (Table 3). Both the drug loading content (DLC) and the drug loading efficiency (DLE) increased with the generation number of copolymers. However, for the same generation, the minor variations observed do not seem to correlate with parameters such as differences in branching density among the samples. Sample G2PBG_15_-*g*-PEO/DOX nevertheless has slightly higher values of DLC (10.8%) and DLE (65%) as compared with the two other arborescent copolymers. Common sense dictates that larger particles should be capable of loading more drugs. However, it can be seen that, despite the lower molecular weight of G2PBG_15_-*g*-PEO and its lower hydrodynamic diameter relative to G2PBG_65_-*g*-PEO, it has slightly higher DLC and DLE values. This could be related to the denser core of G2PBG_15_-*g*-PEO micelles creating a more hydrophobic microenvironment for the drug.

The size and size polydispersity (PDI) of all six DOX-loaded and empty micelles in PBS are compared in Table 3. It can be seen that for each arborescent copolymer, the size measured by DLS before and after drug loading is almost identical. Given that for a 10% drug loading, the diameter of a micelle is only expected to increase by ca. 3%, the very small increases observed seem to be reasonable. As shown in Table 3, the drug loading content (DLC) and the drug loading efficiency (DLE) increased with the generation number of the copolymers, due to the higher PBG content of G2PBG-*g*-PEO/DOX relative to G1PBG-*g*-PEO/DOX. The following release experiments mainly focused on G2PBG-*g*-PEO/DOX systems, which had the highest DLC and DLE.

### 3.5. In Vitro Drug Release Kinetics

An in vitro drug release study was conducted using a dialysis method at 37 °C, in phosphate-buffered saline (PBS), and at two different pH values (7.4 and 5.5). The slightly acidic environment (pH 5.5) was used to simulate the pH of endosomal or lysosomal microenvironments. As shown in Figure 7, the results demonstrate that the pH affected the DOX release rate from all four DOX-loaded unimolecular micelles investigated. The release rate was slower at pH 7.4, with total release after 50 h reaching 12, 14.4, and 19.7% for G2PBG_15_-*g*-PEO/DOX (Figure 7a), G2PBG_29_-*g*-PEO/DOX (Figure 7b) and G2PBG_65_-*g*-PEO/DOX (Figure 7c), respectively, as compared with 17.7, 21.6 and 25.6% at pH 5.5. The relatively slower release at pH 7.4 is attributed to stronger hydrophobic interactions between the core of the nanocarriers and DOX. These results demonstrate that the DOX-loaded unimolecular micelles exhibit pH-dependent in vitro release behavior, whereby faster release of DOX takes place in an acidic environment (e.g., in endosomes/lysosomes) than at physiological pH (e.g., the bloodstream). The protonation of DOX in the acidic environment would also lead to higher aqueous solubility for DOX since this would weaken the hydrophobic interactions between the DOX and the core of the micelles, and thus enhance DOX release [37,38]. This pH-dependent drug release behavior is highly desirable for drug delivery applications, as it should increase the efficacy of cancer therapy while minimizing undesirable side effects [39,40]. For comparison, the release profile for free DOX is also presented (Figure 7e). The sustained release was observed for all the DOX-loaded micelles, in contrast to the burst release seen for free DOX in the absence of the copolymers.

The drug release mechanism from unimolecular arborescent micelles is expected to be passive diffusion. It is generally recognized that the smaller the mean particle diameter, the shorter the diffusion path for drug release, and thereby the higher the release rate. The release rate of DOX at each pH was indeed slower for all the G2PBG-*g*-PEO/DOX systems as compared with G1PBG_29_-*g*-PEO/DOX. For example, the total amount of drug released from G2PBG_29_-*g*-PEO/DOX (Figure 7b) after 50 h reached 14.4 and 21.6% at pH 7.4 and at pH 5.5, respectively, as compared with 30.5 and 34.9% from G1PBG_29_-*g*-PEO/DOX (Figure 7d). This is a consequence of the larger hydrodynamic diameter of G2PBG-*g*-PEO/DOX relative to G1PBG-*g*-PEO/DOX, which increases the diffusion path for drug release, thus reducing the release rate. However, since the G2PBG micelles used in the current investigation, all had comparable diameters, no correlation between the micelle diameter and the release rate was attempted. Another parameter that might influence the release rate of the drug is the structural rigidity of the carrier, i.e., the branching density (*b*_d_). As pointed out earlier, the branching density increased in the series G2PBG_65_ < G2PBG_29_ < G2PBG_15_. Interestingly, the amount of drug released after 50 h was inversely related to the branching density of the PBG cores, namely G2PBG_15_ < G2PBG_29_ < G2PBG_65_. The significant differences in release rate observed within this series of micelles are attributed to their different core *b*_d_. Very compact and dense cores (G2PBG_15_, *b*_d_ = 0.80) would indeed be expected to prevent the diffusion of the drug molecules out of the core more strongly, thus resulting in a slower release. In contrast, a more porous core structure (e.g., G2PBG_65_, *b*_d_ = 0.11) should facilitate the diffusion process. The results obtained therefore strongly suggest that the release rate of DOX can be controlled through variations in the branching density of the core for the randomly grafted arborescent copolymers.

An investigation of drug release over an extended time period (up to 30 days) was also carried out (Appendix A). It revealed that in all cases, the DOX release profiles displayed a biphasic release pattern at both low and high pH, with an initial burst release followed by sustained release of the drug. At neutral pH (7.4) for example, 7% of the loaded DOX was released from G2PBG_65_-*g*-PEO/DOX over the first 6 h and only 19.7% after 50 h. Due to the random grafting of the PEO chains onto the PBG substrate, the burst release is attributed to the drug solubilized in the interfacial region of the micelles, which can diffuse more easily and faster into the release medium. The sustained release behavior of the unimolecular micelles again correlates with the branching density, with a larger cumulative release observed for the micelles with the lowest branching density (G2PBG_65_-*g*-PEO/DOX). Even after one month the release of DOX from the micelles was incomplete (Appendix A). This could be due in part to strong hydrophobic interactions between the PBG core and the DOX molecules, which was indeed confirmed by analysis of the micelles left in the dialysis bag. A significant amount of DOX may also be physically trapped deeper within the hydrophobic PBG core of the micelles, which would make its diffusion out of the highly dense and branched PBG core more difficult. Such a hydrophobic environment can prevent the protonation of DOX, and thus decrease the diffusion ability of DOX. A similar explanation was suggested by Njikang et al. [26] for the incomplete release of indomethacin from highly branched arborescent polystyrene-*g*-poly(2-vinylpyridine) copolymers, serving as non-biocompatible model systems with a structure similar to the unimolecular micelles used in the current investigation.

## 4. Conclusions

A library of water-soluble amphiphilic arborescent copolymers PBG-*g*-PEO was successfully prepared. Their aqueous solution behavior revealed that the copolymers existed mostly as unimolecular (non-aggregated) micellar species. The ability of these unimolecular micelles to encapsulate and release DOX was correlated with the generation number of the copolymers and the branching density (*b*_d_) of the hydrophobic cores. Both the DLC and the DLE increased with the generation number of the copolymers. For each generation, slightly higher values of DLE and DLC were obtained for the micelles with a denser core structure (G1PBG_15_ and G2PBG_15_), with a DLE of about 50% and 65%, respectively, and a DLC of about 8.3% and 10.8%, respectively. Both the DLE and DLC decreased slightly as the grafting density of the cores decreased. The release profiles indicated that the drug release rate could be controlled through the generation number of the copolymers and by modulating the grafting density, and thereby the porosity of the cores. Micelles with highly branched cores (G2PBG_15_) reduced the diffusion rate of the drug, while a lower branching density (G2PBG_65_) facilitated the diffusion process. The versatile encapsulation and release properties of these unimolecular micelles show that they could be useful as nanocarriers for a broad range of drug release applications.

While these unimolecular micelles have already displayed good potential for controlled and sustained drug release, it would be interesting to evaluate the release dynamics over a range of pH values from 5.5–7.4, as well as the cellular uptake of these systems both in vitro and in vivo.

## Data Availability

Data will be made available on request.

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
