# Peer review of "Unimolecular Micelles from Randomly Grafted Arborescent Copolymers with Different Core Branching Densities: Encapsulation of Doxorubicin and In Vitro Release Study"

_materials, 2023, doi:10.3390/ma16062461_

Round 1
Reviewer 1 Report
In this manuscript, Gauthier and Alsehli report the study of unimolecular micelles from randomly grafted arborescent copolymers for encapsulation and in vitro release of Doxorubicin. The copolymers based on poly(γ-benzyl L-glutamate) (PBG) are designed and synthesized, and poly(ethylene oxide) (PEO) is modified randomly in different core branching densities. They all show good encapsulation capability of Doxorubicin, and the release dynamics exhibits faster in pH 5.5 than that in pH 7.4. The experimental data sufficiently support the conclusion, and the manuscript is well-written. I would recommend this manuscript to be published in Materials after addressing some revisions below:
1 In Figure 1, the integrals of all the peaks are suggested to be larger for more clearly exhibition of the ratio of protons.
2 In Scheme 2, the partial hydrolysis for the secondary modification seems not exhibit clearly, please adjust the items for better exhibition.
3 Please give more explanation for the formation of unimolecular micelles instead of multimolecular micelles. Only the data of DLS and morphology seems not so solid.
4 In Figure 6c, the peaks of DOX seem not so clear, please provide a clearer spectrum.
5 Figure 7 shows that the copolymers show potential for controllable release in tumor environment. It is better to study the release dynamics when the pH changes from 5.5 to 7.4 in situ.
6 The authors may consider citing the latest progress on controllable load and release nanoparticles, such as Nat. Commun. 2022, 12, 4993; ACS Appl. Mater. Interfaces, 2018, 10, 24987.
Author Response
In this manuscript, Gauthier and Alsehli report the study of unimolecular micelles from randomly grafted arborescent copolymers for encapsulation and in vitro release of Doxorubicin. The copolymers based on poly(γ-benzyl L-glutamate) (PBG) are designed and synthesized, and poly(ethylene oxide) (PEO) is modified randomly in different core branching densities. They all show good encapsulation capability of Doxorubicin, and the release dynamics exhibits faster in pH 5.5 than that in pH 7.4. The experimental data sufficiently support the conclusion, and the manuscript is well-written. I would recommend this manuscript to be published in Materials after addressing some revisions below:
1 In Figure 1, the integrals of all the peaks are suggested to be larger for more clearly exhibition of the ratio of protons.
The integrals in Figure 1 were renormalized such that the signal for the benzylic protons now has a value of 1.00; the decrease in the integral for the terminal methyl group signal as Xn increases is more obvious using that representation. Larger labels were also used for the integral values.
2 In Scheme 2, the partial hydrolysis for the secondary modification seems not exhibit clearly, please adjust the items for better exhibition.
The scheme has been modified to better illustrate the introduction of carboxyl coupling sites by hydrolysis.
3 Please give more explanation for the formation of unimolecular micelles instead of multimolecular micelles. Only the data of DLS and morphology seems not so solid.
This topic is discussed in the section “Properties of the Arborescent Copolymer Micelles” starting on page 10, line 412, based mainly on the size distribution curves derived from DLS measurements. The size distributions measured for the micelles in a non-selective solvent (DMF, good for both the core and shell portions of the molecules) and in a selective solvent (aqueous phosphate-buffered saline - PBS, non-solvent for the core, good solvent for the shell) are provided in Figure 3, in terms of three different analysis protocols: Distributions on a number basis (essentially equivalent to counting the number of molecules in each fraction of the size distribution), a volume basis (the volume occupied by each fraction in the size distribution), and on an intensity basis (from the intensity of the light scattering signal due to each fraction). Let us focus on the measurements done in the aqueous PBS solutions, since aggregation was clearly not observed in DMF. As discussed in Malm A.V.; Corbett, J. C . W. Sci. Rep. 2029, 9, 13519, https://doi.org/10.1038/s41598-019-50077-4, and multiple other papers dealing with DLS analysis, one of the major limitations of that technique is the fact that the scattering intensity is proportional to the sixth power of the diameter of the particles. Because of that, even a small amount of aggregates present in a sample can cause important distortions (skewing) in intensity-weighted distributions, or even lead to masking of smaller size populations by a tiny amount of large aggregates. This is why number and volume distributions were provided in Figure 3 in addition to intensity distributions. The main point in the comparison is that while significant amounts of aggregates were detected in the intensity distributions (top right plot), which as discussed above, are **extremely** sensitive to aggregates, their presence is almost insignificant in the volume distributions (middle right plot) and non-existent in the number distributions (bottom right plot). This is our basis for stating that the vast majority of the molecules exist in a non-aggregated or unimolecular micelle state. If we were looking at individual molecules and counting a large number of them, we may not see any aggregated species at all, or else extremely few. To point out the issue of DLS data distortion by aggregates, the sentence starting on page 11, line 442 was rewritten as “These results indicate that even though the vast majority of randomly grafted arborescent copolymer molecules exist as stable, isolated unimolecular micelles, a minute amount of aggregation occurs that is only detectable in the intensity-weighted distributions, due to the extremely high sensitivity of the scattering intensity to large particles [34]”, with the reference above cited in support.
4 In Figure 6c, the peaks of DOX seem not so clear, please provide a clearer spectrum.
The whole point of the DOX signals missing in Figure 6c is that the DOX is solubilized in the core of the micelles. Because the micelles are dispersed in D2O, a non-solvent for the PBG component, the core is collapsed into a solid-like state with little to no mobility. Under these conditions, the NMR signal for the DOX component should disappear as seen in Figure 6c, in contrast with DOX in solution (Figure 6a). This is discussed starting on page 13, line 512:
“The complete disappearance of both the PBG and DOX signals for G2PBG15-g-PEO/DOX in D2O (Figure 6c) in comparison to the characteristic signals observed of free DOX in D2O at the same overall DOX concentration (Figure 6a) clearly confirms the restricted mobility of both the PBG and DOX components within the core of G2PBG15-g-PEO. This again confirms the solubilization of DOX within the core of the micelles in aqueous solutions and the successful encapsulation of the DOX molecules’’.
5 Figure 7 shows that the copolymers show potential for controllable release in tumor environment. It is better to study the release dynamics when the pH changes from 5.5 to 7.4 in situ.
We thank the reviewer for the suggestion, as the release behavior would also be important to investigate over the whole pH 5.5-7.4 range. This would obviously involve a large number of experiments, which fall outside the scope of this initial investigation. In recognition of this, the paragraph below was added in the Conclusions section starting on page 17, line 640.
While these unimolecular micelles have already displayed good potential for controlled and sustained drug release, it would obviously be interesting to evaluate the release dynamics over a range of pH values from 5.5-7.4, as well as the cellular uptake of these systems both in vitro and in vivo.
6 The authors may consider citing the latest progress on controllable load and release nanoparticles, such as Nat. Commun. 2022, 12, 4993; ACS Appl. Mater. Interfaces, 2018, 10, 24987.
The reference ACS Appl. Mater. Interfaces, 2018, 10, 24987 is now cited on page 15, line 568. The other reference provided seems unrelated to drug release, unless there was a mistake in it.
Reviewer 2 Report
I have read the manuscript “Unimolecular Micelles from Randomly Grafted Arborescent Copolymers with Different Core Branching Densities: Encapsulation of Doxorubicin and In Vitro Release Study” submitted to Materials MDPI
The subject of this manuscript is interesting and the authors assembled valuable information on new strategy in the developing of drug delivery nanocarriers by biocompatible amphiphilic arborescent copolymers synthesized by ring-opening polymerization. The document is comprehensive, the discussion is reasonable, meanwhile, there are some missing data and minor corrections/clarifications need to be made for the current document. So, I think this document should be considered for publication in Materials only after proper minor modifications. Some of my specific comments are below:
Point 1: Abstract. For the sentence “These amphiphilic copolymers behaved like unimolecular micelles in aqueous solutions in most cases.” Avoid generalizations. Additional explication is necessary, please indicate what is the implication of this statement.
Point 2: Abstract. For the sentence “It was demonstrated that these biocompatible copolymers can encapsulate … with high efficiency. Avoid generalizations. Additional explication is necessary, please indicate against which value this efficiency value should be compared for a better understanding of this sentence.
Point 3. 2.1. Materials. Please indicate for DMF purified by distillation under reduced pressure and for the other purified reagents. What is the level of purity required and what was achieved in this work?
Point 4. 2.2. Characterization section, I recommend to authors briefly indicate in this section the conditions and signals of interest for NMR analysis.
Point 5. 2.2. Characterization section, please indicate what stands for SEC and GPC for better clarity.
Point 6. 2.3. Synthetic Procedures, please indicate that the polymerization technique applied was ROP.
Point 7. Conversion/yield of polymerizations should be included.
Point 8. Polydispersity should be read as dispersity. PDI as Ð= Mw/Mn.
Point 9. Why is it important to keep low dispersion values of the copolymers in this work?
Point 10. Table 1. The thousand separators must be used for molar mass data.
Point 11. The authors mentioned that “ Unfortunately, TEM and AFM imaging indicated that the molecules were all spherical” … Additional explication is necessary.
Point 12: The authors should make a more critical or detailed comparison of their work with other literature reports.
Author Response
I have read the manuscript “Unimolecular Micelles from Randomly Grafted Arborescent Copolymers with Different Core Branching Densities: Encapsulation of Doxorubicin and In Vitro Release Study” submitted to Materials MDPI
The subject of this manuscript is interesting and the authors assembled valuable information on new strategy in the developing of drug delivery nanocarriers by biocompatible amphiphilic arborescent copolymers synthesized by ring-opening polymerization. The document is comprehensive, the discussion is reasonable, meanwhile, there are some missing data and minor corrections/clarifications need to be made for the current document. So, I think this document should be considered for publication in Materials only after proper minor modifications. Some of my specific comments are below:
Point 1: Abstract. For the sentence “These amphiphilic copolymers behaved like unimolecular micelles in aqueous solutions in most cases.” Avoid generalizations. Additional explication is necessary, please indicate what is the implication of this statement.
The sentence has been rewritten to specify that no significant amounts of aggregates were detected in number- and volume-weighted size distributions derived from DLS measurements. The sentence starting on page 1, line 22 now reads “These amphiphilic copolymers behaved like unimolecular micelles in aqueous solutions, with unimodal number- and volume-weighted size distributions in dynamic light scattering measurements.”
Point 2: Abstract. For the sentence “It was demonstrated that these biocompatible copolymers can encapsulate … with high efficiency. Avoid generalizations. Additional explication is necessary, please indicate against which value this efficiency value should be compared for a better understanding of this sentence.
The above sentence has been modified to “It was demonstrated that these biocompatible copolymers can encapsulate hydrophobic drugs such as doxorubicin (DOX) within their hydrophobic core with drug loading efficiencies of 42-65%” (page 1, line 24) to be more specific.
Point 3. 2.1. Materials. Please indicate for DMF purified by distillation under reduced pressure and for the other purified reagents. What is the level of purity required and what was achieved in this work?
The DMF used was purified and stored in the dark as described previously [27,28], but no tests were carried out to determine the purity of the solvent. The presence of impurities such as cyanide anions, generated photochemically in DMF, can decrease the amine functionality level of the chain ends, thereby affecting the grafting yield in the coupling reaction and the uniformity (dispersity) of the PBG serving as side chains [27,28]. These references were added on page 3, line 101 for readers potentially interested in these synthetic details.
Point 4. 2.2. Characterization section, I recommend to authors briefly indicate in this section the conditions and signals of interest for NMR analysis.
Specific signals of interest are provided in the Results and Discussion section, in addition to the Supporting Information (SI). For example, the determination of the number-average degree of polymerization (Xn) by NMR analysis is described on page 6, line 270. The determination of the deprotection level by NMR analysis is discussed starting on page 7, line 289. The use of 19F NMR analysis to determine the functionality level of the PGB serving as side chains is described in the SI.
Point 5. 2.2. Characterization section, please indicate what stands for SEC and GPC for better clarity.
We apologize for the confusion. The abbreviations SEC (size exclusion chromatography, already defined on page 2, line 91) and GPC (gel permeation chromatography) are equivalent and interchangeable. The GPC abbreviation used on page 3, line 125 has been changed to SEC. The other occurrence of GPC (page 3, line 124) cannot be changed because it is part of the model name of the instrument used (Viscotek GPCmax).
Point 6. 2.3. Synthetic Procedures, please indicate that the polymerization technique applied was ROP.
The fact that the polymerization technique used was ring-opening polymerization is now specified on page 4, line 167.
Point 7. Conversion/yield of polymerizations should be included.
The yield of the reactions was included in Table 1 and in the Supporting Information (SI). More specific references to the SI have been added on page 4, line 176 and page 5, line 200 regarding the yield.
Point 8. Polydispersity should be read as dispersity. PDI as Ð= Mw/Mn.
References to the PDI in relation to the SEC measurements have been changed to dispersity Ð.
Point 9. Why is it important to keep low dispersion values of the copolymers in this work?
The sentences below were added on page 2, line 62 to explain the desirability of a uniform particle size distribution for drug delivery applications, with a reference in support:
The ability of nanocarriers to accumulate in target tissues depends on their physicochemical characteristics, among which the particle size is critical. The formulation of safe, stable and efficient drug delivery systems, with well-defined and predictable release characteristics, requires nanocarriers with a uniform (close to monodisperse) size population [24].
Point 10. Table 1. The thousand separators must be used for molar mass data.
Commas have been added as thousand separators for the Table 1 entries.
Point 11. The authors mentioned that “ Unfortunately, TEM and AFM imaging indicated that the molecules were all spherical” … Additional explication is necessary.
The paragraph below has been added on page 6, line 262 to explain the interest in the shape of nanoparticles serving in drug encapsulation, with references in support:
The shape of micelles can significantly alter their behavior in living organisms and cells. For example, it was reported that worm-like polymer micelles of amphiphilic poly(ethylene glycol)-block-polycaprolactone copolymers had an increased circulation time in comparison with their spherical counterparts [32,33].
Also on page 6, line 262, the text “…from spherical to brush-like topologies” was added.
Point 12: The authors should make a more critical or detailed comparison of their work with other literature reports.
Other literature reports related to the current investigation have already been discussed in the Introduction.